# Self-driving microscopy detects the onset of protein aggregation and enables intelligent Brillouin imaging

Khalid A. Ibrahim [1,2,3] ✉, Camille Cathala [1], Carlo Bevilacqua [4], Lely Feletti [1,3], Robert Prevedel [4,5], Hilal A. Lashuel[2] ✉ & Aleksandra Radenovic [1,3] ✉

The process of protein aggregation, central to neurodegenerative diseases like Huntington's, is challenging to study due to its unpredictable nature and relatively rapid kinetics. Understanding its biomechanics is crucial for unraveling its role in disease progression and cellular toxicity. Brillouin microscopy offers unique advantages for studying biomechanical properties, yet is limited by slow imaging speed, complicating its use for rapid and dynamic processes like protein aggregation. To overcome these limitations, we developed a self-driving microscope that uses deep learning to predict the onset of aggregation from a single fluorescence image of soluble protein, achieving 91% accuracy. The system triggers optimized multimodal imaging when aggregation is imminent, enabling intelligent Brillouin microscopy of this dynamic biomechanical process. Furthermore, we demonstrate that by detecting mature aggregates in real time using brightfield images and a neural network, Brillouin microscopy can be used to study their biomechanical properties without the need for fluorescence labeling, minimizing phototoxicity and preserving sample health. This autonomous microscopy approach advances the study of aggregation kinetics and biomechanics in living cells, offering a powerful tool for investigating the role of protein misfolding and aggregation in neurodegeneration.

Protein aggregation is the hallmark of various neurodegenerative diseases (NDDs), including Huntington's disease (HD), which is caused by an abnormal polyglutamine (polyQ) expansion in the first exon of the Huntingtin (Htt) protein (Httex1), leading to its increased propensity to misfold and aggregate[1–4]. Our understanding of the mechanisms and kinetics of the aggregation process and its role in the pathogenesis of HD and other NDDs remains incomplete. Whether protein aggregation and inclusion formation are the causes or consequences of neurodegeneration, and whether they represent neuroprotective mechanisms aimed at sequestering misfolded and toxic proteins, remains unclear[5–9]. It has also been hypothesized that the oligomeric species that are formed in an early stage of the process could have a major role in in the initiation or progression of the cascade of events leading to neurodegeneration[7].

The transition from soluble Httex1 protein to fully-mature aggregate has been observed to be relatively rapid, in the order of a few minutes to a few tens oof minutes, compared to the large time window of days that is usually required for the formation of protein

[1]Laboratory of Nanoscale Biology, École Polytechnique Fédérale de Lausanne (EPFL), Lausanne, Switzerland. [2]Laboratory of Molecular and Chemical Biology of Neurodegeneration, École Polytechnique Fédérale de Lausanne (EPFL), Lausanne, Switzerland. [3]NCCR Bio-Inspired Materials, École Polytechnique Fédérale de Lausanne (EPFL), Lausanne, Switzerland. [4]Cell Biology and Biophysics Unit, European Molecular Biology Laboratory (EMBL), Heidelberg, Germany. [5]German Center for Lung Research (DZL), Heidelberg, Germany. ✉e-mail: khalid.mohie@gmail.com; hilal.lashuel@epfl.ch; aleksandra.radenovic@epfl.ch

**Fig. 1 | Self-driving microscopy predicts the onset of protein aggregation. a** The process of protein aggregation occurs unpredictably in different locations at different times, over a large time window of 48 h, yet the transition from soluble protein to mature aggregate is stochastic and relatively rapid (few minutes–few tens of minutes), making it difficult to capture optimally. Images are representative of experiments conducted more than three independent times. Scale bar: 200 μm. **b** Self-driving microscopy is an approach that uses a feedback loop to autonomously adjust the imaging settings once an event is detected by a real-time model, offering advantages such as optimal use of time and computing resources. **c** In standard microscopy, images are acquired and then later processed and analyzed. In our SDM approach, processing and computation are done in real-time by a neural network model, AEGON, that was previously trained. This model is capable of detecting the onset of protein aggregation, triggering an optimized acquisition to ideally capture the process. Images are representative of experiments conducted more than three independent times. Scale bars: 5 μm.

fibrils and inclusions (Fig. 1a)[10]. Aggregation occurs stochastically in different parts of the sample at different times, complicating our ability to study early steps in the process. Fluorescence time-lapse microscopy and image-based high-throughput screening are often used to monitor the aggregation process over many hours[10]. This is made possible by the high field of view (FOV), which is further increased by stitching. However, this comes at the cost of low magnification and resolution, and limited temporal resolution, depending on the size of the chosen region of interest (ROI). The unpredictability and fleeting nature of this phenomenon call for new tools that are able to capture the process with optimal spatiotemporal resolution and specificity.

State-of-the-art microscopy modalities are indispensable tools that have significantly advanced our understanding of biological systems at various scales. However, all microscopy techniques inherently involve trade-offs between key parameters, such as the spatial resolution, temporal resolution, light dose, penetration depth, and signal-to-noise ratio (SNR)[11–13], depending on their underlying principles of operation. The development of integrative approaches that leverage the strengths of complementary techniques using an intelligent microscopy strategy could help address these limitations and enable the investigation of complex and dynamic biological processes. Correlative microscopy techniques[14] and multimodal microscopes[15,16] have been developed to combine different imaging modalities, but this usually involves complex workflows and is typically not autonomous or adaptive in real-time. Autonomous microscopy approaches have recently started to emerge; in such approaches, there is a feedback loop between the data that is being acquired, a computer that processes this real-time data, and the microscope control software (Fig. 1b)[13,17,18]. This allows the microscope to compute and infer the current needs—whether in terms of illumination laser power, exposure time, spatial resolution, or any other such parameter, and adapt the imaging settings accordingly. The advantages of this method include preserving sample integrity, saving time and data resources, and moving closer to fully automated microscopy capable of capturing transient biological processes. Examples include autonomous control of the acquisition settings[19–21] and adaptive illumination/detection to reduce photobleaching and phototoxicity[22–27], to speed up the acquisition, or to enhance the image quality. Recently, event-triggered approaches have been implemented, where the acquisition rate or imaging modality is adjusted based on the detection of specific events[28–30]. To date, most implementations of this approach have focused on switching between fluorescence microscopy

techniques. However, switching to a label-free, low-light dose technique would result in minimizing phototoxicity and light exposure, offering significant benefits for long-term live-cell imaging. Self-driving microscopy (SDM) holds considerable untapped potential for optimally capturing the dynamics and kinetics of cellular and molecular processes, such as protein aggregation, in real-time, particularly in intact, living systems.

Recently, Brillouin microscopy (BM) has emerged as a non-invasive, label-free tool to study the biomechanical properties of cells and tissues, which are critical for understanding cellular behavior and tissue development[31–34]. BM is based on inelastic Brillouin scattering, which provides information about the longitudinal modulus, a measure of viscoelasticity, allowing insight into the high-frequency mechanical properties of the specimen[33]. Recently, BM was used to gain insight into the mechanical properties of stress granules[35] and enabled new insight into Httex1 aggregates[36], revealing that they exhibit a higher longitudinal modulus compared to the cytoplasm. The relationship between the mechanical properties of these aggregates and their dynamic properties remains unexplored. BM has only been used to image mature aggregates, rather than to monitor and characterize the full process of aggregation, due to the slow speed of the instrument, which arises from the need for point-by-point acquisition in conventional BM instruments[33,35–37]. Each measurement involves scanning a small region of the sample and analyzing the frequency shift of scattered light, which is time-consuming. Additionally, the low scattering efficiency of Brillouin signals necessitates long acquisition times to achieve a sufficient SNR, further limiting the ability to capture dynamic processes. Line-scanning and nonlinear modalities have been developed to increase the speed of this imaging technique, but they come with their own challenges, including complex sample mounting requirements and potential sample perturbation[33]. Alternative tools for studying biomechanics, such as atomic force microscopy (AFM)[38,39] or optical tweezers (OT)[40], face similar limitations with regard to speed, and in particular suffer from being invasive, further highlighting the need for more refined techniques[41].

To address many of the limitations highlighted above, we have developed a self-driving, fully-modular microscope that enables capturing the different stages of protein aggregation and enables intelligent Brillouin imaging (Fig. 1c). Compared to standard microscopy, which can be automated to acquire multiple FOVs that are to be processed and analyzed post-acquisition, our self-driving microscope processes the images in real time and uses a deep learning (DL) model to predict if aggregation is about to occur. When an aggregation event is imminent, the system initiates an optimized multimodal acquisition, capturing the process with high specificity and spatiotemporal resolution. Our model for predicting aggregation onset (AEGON) achieves 91% test set accuracy at predicting aggregation purely from images of soluble protein, a task that has been challenging to achieve with human vision. We compare the performance of different state-of-the-art neural network architectures on this task and demonstrate the high potential of vision transformers (ViTs). By generalizing our classification model to one-plane, one-time-point fluorescence image inputs and showcasing its functionality on a completely different setup, we provide a simple-to-use method for automatically and optimally capturing aggregation kinetics and dynamics. We apply our SDM pipeline to address two significant challenges in the field of Brillouin microscopy. First, using our aggregation classification model, we enable dynamic and highly optimized retrieval of the biomechanical properties throughout the aggregation process. Second, we employ another DL model, IC-LINA, to classify whether FOVs contain aggregates based solely on brightfield images, eliminating the need for fluorescent labeling to detect aggregates. This preserves the label-free advantage of BM, facilitating long-term imaging while maintaining sample integrity and viability, and paving the way for future clinical applications using BM for diagnosis and therapeutics. This work advances the potential of self-driving microscopy for capturing complex biological processes in real time, opening new possibilities for tracking protein aggregation dynamics, kinetics, and biomechanics throughout the process and optimally utilizing label-free techniques for enhanced sample preservation.

## Results

### Accurate prediction of protein aggregation triggers optimized multimodal imaging

Determining the onset of protein aggregation is challenging for humans to achieve by visually inspecting images of soluble protein, as there are no apparent discernible features for proteins that are about to aggregate versus those that are not. We aimed to test whether this challenging task can be solved with the use of a neural network. To aid the neural network in learning to identify the onset of protein aggregation and to produce a robust and reproducible model, a large enough dataset should be collected that is representative of the variability of this process, while eliminating sources of error that could confuse the model during training[42]. We used our custom-built microscope (Supplementary Fig. 1) that can capture images in fluorescence, brightfield, and quantitative phase imaging (QPI) and provide us with images at 8 different z-planes simultaneously, using an image-splitting prism and two cameras[43]. We chose a well-characterized cellular model of HD that allows the dynamic study of the process using live-cell imaging[44]. In this model, aggregates form within 48 h when mutant Httex1 with a polyQ repeat length ≥ 39 is overexpressed in HEK293 cells. In our case, we used a construct with a polyQ repeat length of 72, fused to GFP (Httex1-72Q-GFP) to be able to monitor the proteins throughout the aggregation process.

Using our microscope, we collected time-lapse, multi-plane fluorescence images of events and non-events to use for training a classification model, AEGON. The full dataset consists of 71 events and 68 non-events. Events are defined as sequences of images where the proteins ended up aggregating during our acquisition, and non-events are sequences where the proteins did not end up aggregating during our acquisition. Due to the high dynamism of the process, we first trained a combination of a convolutional neural network (CNN) and recurrent neural network (RNN), namely a VGG16 long short-term memory (VGG16-LSTM) network, to make use of the temporal information while maintaining efficiency. However, we eventually found the ViT network to work better and achieved 91% accuracy at predicting the onset of aggregation on our test set, a set of examples that the model had never seen before. Figure 2a shows the classification matrix on our test set, illustrating the high performance of the AEGON model using different standard performance metrics (precision = 1.0, recall = 0.82, F1 score = 0.9, and accuracy = 0.91). Visual test-set examples are provided in Fig. 2b, showing examples of both correctly predicted events and correctly predicted non-events. These examples illustrate the similarity of events and non-events to the human eye and hence the difficulty of visually determining when aggregation is about to occur. The FOV of the input images is ≈39 μm × 39 μm. Generally, the soluble protein aggregates into only one spot during the process. If an FOV contains multiple cells, it is possible that each cell could form a separate aggregate; however, this rarely occurs as the likelihood of two cells forming aggregates at the same time and place is low. During our collection of training events, we aimed to collect a diverse dataset that is well-representative of the different phenomena that could occur and did not exclude any particular kinds of aggregation. The model should, therefore, be able to capture varying types of aggregation events. Further fine-tuning of the model is also possible using transfer learning.

We incorporated AEGON into an SDM pipeline that we developed. In this pipeline, an initial scan is done, after which the model computes whether events are about to occur, and then the microscope autonomously configures the imaging settings and initiates the optimized

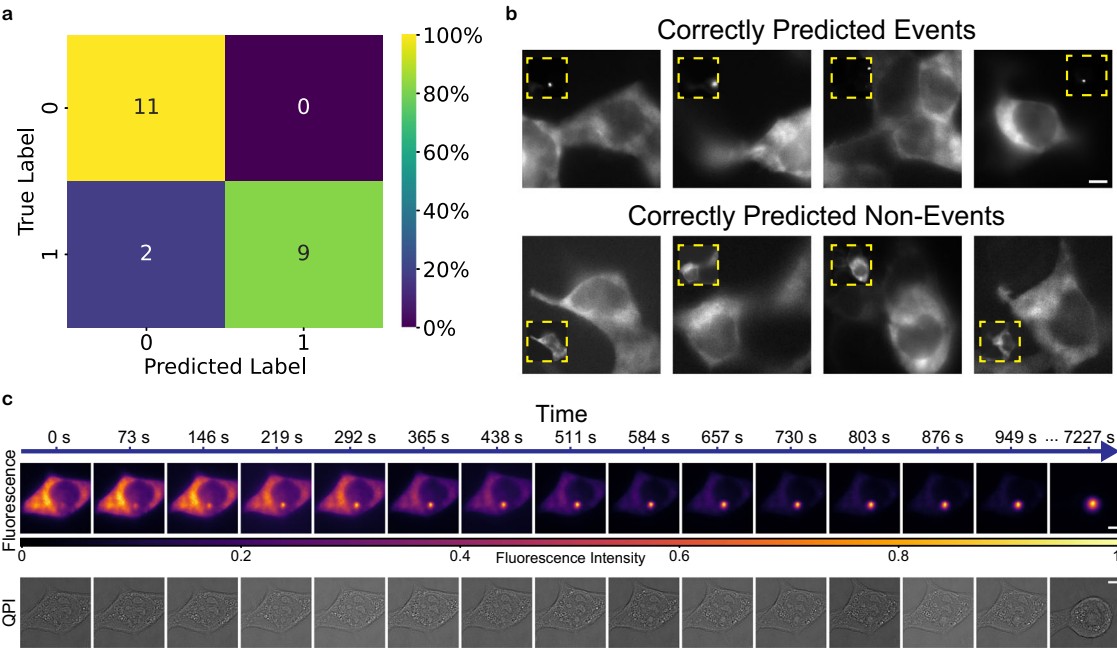

**Fig. 2 | Optimized multimodal imaging triggered by our model, AEGON.**
**a** Classification matrix of our model AEGON for predicting aggregation onset. Performance metrics: precision = 1.0, recall = 0.82, F1 score = 0.9, accuracy = 0.91. **b** Test-set examples of correctly predicted events and correctly predicted non-events. Inset images show the eventual fate of these fields of view. Training and test set data were collected from more than three independent experiments. **c** Multimodal (fluorescence and QPI) images captured after an event is detected by our model within our SDM pipeline. The intensity in each image is locally rescaled between 0 and 1. Images are representative of experiments conducted more than three independent times. Scale bars: 5 μm.

scan. The software is fully modular, enabling the user to set various parameters for both the initial and optimized scans, such as the imaging modalities, exposure times, and laser powers. After the initial scan, images are given in real time as input to the model, which predicts whether aggregation is about to occur. There are two modes, one where a specific acquisition sequence (a patterned scan in both the x and y directions) is performed and all the images are passed to the model together at the end, and one where the images at each FOV are passed directly to the model. In the first mode, all FOVs are checked at the same time, and if there are any aggregation events predicted, they are added to a list, which is iterated over in the optimized scan. In the other mode, once an event is predicted, this directly initiates the optimized scan at that FOV, after which the initial scan continues to check the subsequent FOVs.

With our self-driving microscope, we are able to accurately predict the onset of mutant Httex1 aggregation process and capture it with high spatiotemporal resolution in multiple modalities (Fig. 2c). Here, we show a sequence of images captured using fluorescence microscopy and QPI after AEGON correctly predicted an event in real time. The concept is applicable to any other modalities that are available to the user on the same setup.

### Generalizability of the method
We initially trained a network based on a VGG16-LSTM (Supplementary Fig. 2a) architecture due to the high dynamism of the process we are studying and the high applicability of LSTM networks for temporally-connected data. With similar rationale, we also trained a video vision transformer (ViViT) (Supplementary Fig. 2b). Both architectures produced highly accurate models (88% and 93%). However, we wanted to produce the most easily applicable model for new users in the community. Therefore, we evaluated whether it is possible to achieve similar accuracy using images from one single time-point rather than a sequence of images. We were able to train a ViT model with one time-point, and 8 z-plane inputs, and achieve 91% accuracy, similar to what we could achieve with higher time-points (Fig. 3a).

We then sought to assess whether it is possible to achieve similar accuracy with a one-plane, one-time-point fluorescence input, producing an AEGON model which is even simpler to use. Again, we found that one-plane inputs lead to high accuracy (86%) (Fig. 3b). This is the most generalizable and easy-to-use model, which only requires one image to perform the classification. The input is taken using a widefield fluorescence microscope, which is widely available in laboratories around the world.

Next, to translate this to mechanical (Brillouin) imaging and to show the generalizability of our approach, we implemented it on a separate, custom-built BM setup (Methods), with an entirely different illumination, microscope objective, pixel size, and detector. This setup is capable of brightfield, fluorescence, and Brillouin microscopy. We developed a similar SDM pipeline as in our primary setup, described above, which enables the user to set the parameters for an initial scan and an optimized scan that is triggered when an event is found. Interestingly, we found that AEGON works directly on this different setup, without the need for transfer learning (Fig. 3c), as it was able to correctly predict events and non-events. The FOVs in the images are 35 μm × 35 μm (with a step size of 1 μm for BM images).

### Dynamic Brillouin microscopy of Httex1 aggregation
Monitoring changes in the biomechanical properties throughout the process of protein aggregation has so far been challenging due to the limitations of the instruments that are used in these experiments. Although the emerging BM offers several advantages over standard techniques like AFM and OT, such as being noninvasive and label-free, one of its main limitations is its slow speed due to the weak Brillouin scattering cross-section that entails long data-acquisition times per image pixel. This has hindered the dynamic capture of the viscoelastic properties during aggregation. With a typical pixel dwell time of around 20–50 ms, one confocal-based BM acquisition can take several minutes to hours, depending on the chosen FOV and resolution. This is not a significant issue for fully formed, mature aggregates, which are relatively stable and can be identified using their fluorescence signal.

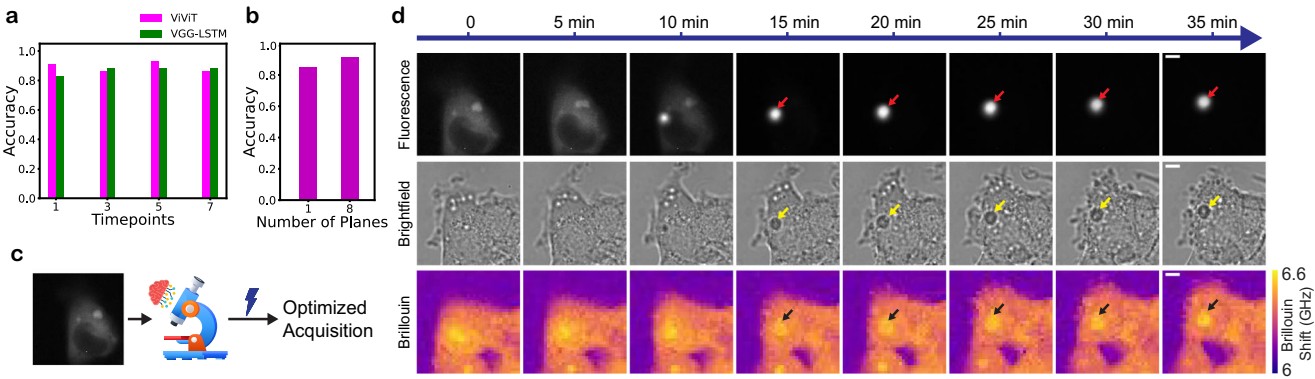

**Fig. 3 | Generalizability of the method enables dynamic Brillouin microscopy.**
**a** Accuracy of the AEGON models we trained for various time-point inputs. One-time-point models maintain high accuracy. **b** Accuracy of our one-time-point ViT model with different z-plane inputs. The one-plane model maintains high accuracy. **c** With our one-plane, one-time-point model, only one fluorescence image is needed to predict protein aggregation. We are able to use our model on a completely different setup and adopt our SDM pipeline, without transfer learning.
**d** Multimodal imaging on a different setup, triggered by our AEGON model, enabling dynamic Brillouin microscopy of the aggregation process. Arrows mark the aggregating region (different colors only serve to optimize the contrast). Images are representative of experiments conducted more than three independent times. Scale bars: 5 μm.

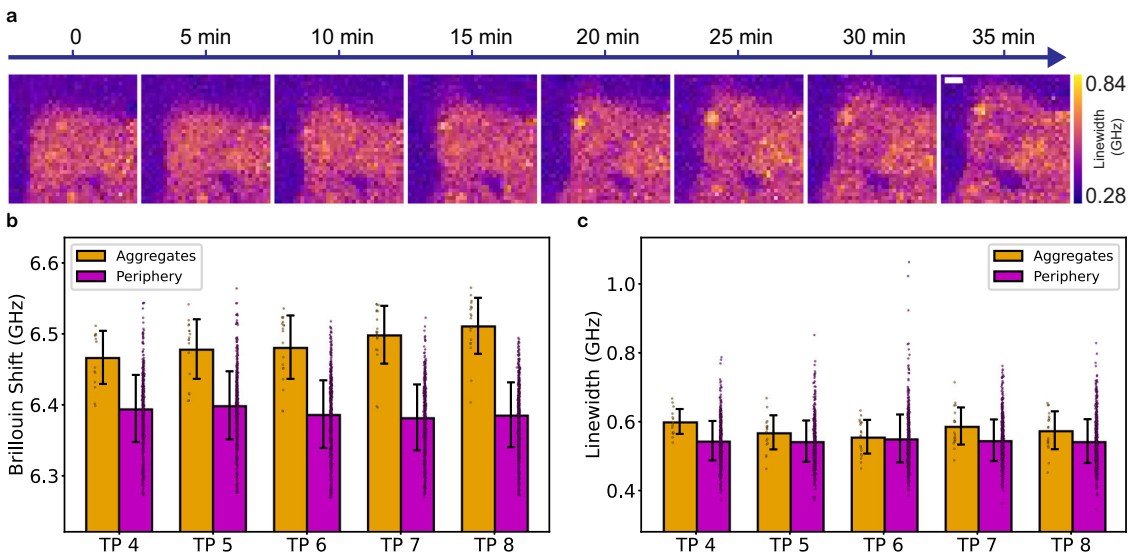

**Fig. 4 | Evolution and comparison of the Brillouin shift and linewidth over time.**
**a** Dynamic imaging of the linewidths throughout aggregation is enabled by the model, which predicts the onset of the process. Scale bar: 5 μm. **b**, **c** Comparison of the Brillouin shifts and linewidths of the aggregating region versus the periphery. TP–timepoint. Data are presented as mean values ± standard deviation, within the respective region in the image, at each timepoint for the shown example. **b** Brillouin shift comparison. **c** Linewidth comparison.

However, the unpredictability of when and where soluble proteins will aggregate makes it extremely challenging to predict early aggregation events and target the right FOV at the right time.

Our SDM pipeline overcomes this challenge and enables us to scan for potential events, only triggering BM when the AEGON model predicts the onset of aggregation. With this strategy, we are able to capture the aggregation process as it occurs with BM (Fig. 3d). We also capture the process with brightfield and fluorescence microscopy. Figure 4 shows the linewidth images throughout the aggregation process and a comparison of the Brillouin shifts and linewidths of the aggregating region and the surrounding cellular periphery across different timepoints. We observe that the region where the aggregate is forming (marked in Fig. 3d) exhibits a higher Brillouin shift compared to the peripheral cytoplasm (the full signal in orange, excluding the background), in agreement with prior studies[36]. This is also true for the linewidths, which are increased in the aggregating region compared to the periphery. We also observe an increasing trend in the Brillouin shift of the aggregating region over time, suggesting an increased stiffness.

A final Brillouin shift of 6.51 ± 0.04 GHz and a linewidth of 0.58 ± 0.06 GHz are measured for the aggregating region, versus a shift of 6.39 ± 0.05 GHz and a linewidth of 0.54 ± 0.06 GHz for the periphery. Our method demonstrates, for the first time to our knowledge, the ability to study the biomechanics of Httex1 aggregation as the process occurs, opening up various possibilities to understand and target different stages of protein aggregation.

**Intelligent Brillouin microscopy enables completely label-free biomechanical studies**

The previously described SDM approach is highly modular and dynamic, as not only can various parameters be set by the user for both the initial and optimized scans, but also the model that performs the classification can be simply chosen by the user. In our first use case, we showed the ability to capture the biomechanical properties of the process as it occurs. Next, we leveraged the flexible and modular nature of our method and extended our intelligent Brillouin microscope to solve another critical issue. Despite BM being a label-free

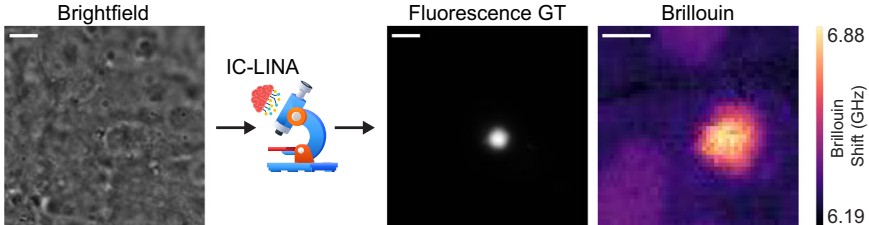

**Fig. 5 | Real-time identification and biomechanical analysis of aggregates using SDM and IC-LINA.** The IC-LINA model detects in real time the presence of aggregates solely from label-free brightfield images, eliminating the need for fluorescence imaging to localize aggregates. Here, the fluorescence image serves as a ground truth to validate aggregate presence. Autonomously-triggered Brillouin microscopy enables detailed biomechanical analysis at identified locations. Images are representative of experiments conducted more than three independent times. Scale bars: 5 μm.

method, fluorescence microscopy is typically needed to identify and localize the aggregates. This introduces unnecessary perturbations and phototoxicity to the sample[44–47]. To solve this issue, we extended our previous work where we developed CNN models for label-free identification of NDD-associated aggregates (LINA)[48]. We developed an image-classification LINA (IC-LINA) model, which takes a single brightfield image as input and determines the presence or absence of aggregates in the image.

IC-LINA is 97% accurate on our test set (Supplementary Fig. 3). We used this model within our SDM pipeline so that the microscope can autonomously identify—in real-time—locations in the sample where aggregates are located, without the need for fluorescence labeling. BM is then only triggered on FOVs where aggregates are identified. Figure 5 shows an example of the results of using our model on a label-free brightfield image, which was correctly classified as having an aggregate, as validated by checking the fluorescence signal as the ground truth, and the corresponding BM image. This real-time, correlative approach enables fully label-free and highly specific biomechanical studies of Httex1 aggregates. This also demonstrates how intelligent microscopy can be used to switch from completely label-free, low-light-dose methods like brightfield microscopy to less gentle techniques like BM and fluorescence microscopy. This reduces the light dose and phototoxicity brought upon the sample, improving the health and longevity of living specimens. The FOV of the input image (and GT fluorescence images) is ≈39 μm × 39 μm, and the Brillouin image focuses on the aggregate region (FOV = 10 μm × 10 μm with a step size of 0.2 μm).

These results highlight the method's high generalizability and ease of use for researchers who want to use it in their own laboratory to study Httex1 aggregation. The method can also be extended to other proteins, whether by transfer learning and fine-tuning this model on a different system or training a new model from scratch using the same strategy and architecture.

While this work focuses on applying SDM and intelligent BM to protein aggregation, the underlying strategy of real-time event detection and adaptive multimodal imaging is broadly applicable to a variety of dynamic cellular processes. For instance, many high-content screening assays rely on protein re-localization events—such as nuclear translocation of transcription factors, signaling molecules, or stress response proteins—as key readouts. Our SDM framework, which can be trained to detect subtle changes from single-plane fluorescence or brightfield images, could be adapted to identify such re-localization events, including nuclear accumulation, cytoplasmic clearance, or organelle-specific targeting. This would enable dynamic switching to higher-resolution or complementary modalities (e.g., super-resolution or label-free techniques), or optimizing other imaging settings such as the temporal resolution or the exposure time/laser power, at the right time and location, enhancing the informativeness of such assays while minimizing phototoxicity and enhancing throughput. We envision that retraining the model on datasets relevant to nuclear import/export, mitotic events, or phase transitions could allow for real-time monitoring of diverse biological phenomena across cell biology, developmental systems, or drug screening applications.

## Discussion

We developed a method for self-driving microscopy that can detect the onset of protein aggregation and capture this unpredictable process in optimized spatiotemporal resolution, enabling the microscopist to precisely determine and distinguish the parameters for exploration and acquisition. Our DL-based AEGON model is 91% accurate at this challenging task and can differentiate between highly similar images of soluble proteins to decipher when aggregation will occur. We show the results of our autonomous, real-time approach, capturing the process with multiple modalities, including BM. This technique also enables, for the first time, the retrieval of biomechanical properties throughout the process. The method is highly generalizable and can easily be adopted on other setups, as we showcased by developing a similar SDM pipeline and running the same model on a completely different setup, without requiring transfer learning. One fluorescence image of the protein in the soluble state is all that is needed to make the classification. Furthermore, other models can be used to classify different kinds of events of interest. We demonstrate this by developing an image classification model, IC-LINA, that can identify aggregates from label-free, brightfield images and correlating them with BM in real time, solving a critical issue of relying on fluorescence to identify FOVs that contain aggregates. Our method also demonstrates the ability to switch modalities from a label-free technique, such as brightfield, instead of relying on a fluorescence-based technique, thereby better preserving the sample integrity. Identifying mature aggregates in a label-free manner and dynamically capturing their properties have remained unsolved challenges until now. We envisage great potential for our tools and their application to address these challenges and provide critical insights into the pathophysiological process underlying aggregation. Incorporating additional label-free techniques such as Raman microscopy—which has previously been correlated with BM[49–51]—to obtain real-time information regarding the biochemical and structural composition of aggregates and intelligently switching between the different techniques can open new avenues for self-driving, non-invasive microscopy.

The field of deep learning is undergoing rapid and continuous evolution, with new advancements and architectures emerging at an unprecedented pace. We assessed the performance of multiple deep learning architectures, specifically VGG16, VGG-LSTM, and ViT. Our results show that, while the different architectures all lead to good performance, our ViT models outperform those of the other two. This highlights the potential of the ongoing advancements in model design and the importance of selecting the right architecture for specific tasks. ViT models leverage self-attention mechanisms that allow them to capture intricate relationships in the data more effectively than

traditional CNNs. This capability appears to provide a significant advantage in predicting the onset of aggregation.

BM provides insight into the high-frequency longitudinal modulus of biological materials and is therefore sensitive to their viscoelastic properties. Prior work has shown that Brillouin shifts increase in more solid-like structures, such as mature Httex1 aggregates, compared to the surrounding cytoplasm[36]. This has also been shown for biomolecular condensates formed via liquid-liquid phase separation (LLPS), such as stress granules that are formed through the expression of FUS and the presence of oxidative stress[35]. Recent work uses BM as a measure for the average molecular interaction and finds that conditions that favor LLPS increase the Brillouin shift, linewidth, and protein concentration[52]. We expect that the stiffness of aggregating proteins will increase as they transition from liquid-like states to the solid-like mature-aggregate state, due to higher protein density and stronger intermolecular interactions that reduce mobility and increase mechanical rigidity. Our SDM-Brillouin framework is well suited for exploring a continuum of material states throughout the aggregation process, including transient, dynamic condensates, as it enables dynamic imaging throughout the process for the first time, and as shown here, we observe an increase in the Brillouin shift as the aggregate forms. Future work, which incorporates additional modalities, such as fluorescence recovery after photobleaching (FRAP), may allow for a more systematic correlation between internal dynamics and Brillouin mechanical signatures, thus enabling new insights into the phase behavior and maturation of aggregates in neurodegenerative disease models.

Growing evidence suggests that the process of inclusion formation involves a complex interplay between misfolded protein aggregates, lipids, membranous organelles, and other subcellular constituents[44,53]. These include cytoskeletal proteins, which provide the structural framework that allows cells to maintain their shape, resist deformation, generate mechanical forces, and respond to mechanical stress, underscoring the importance of developing tools that are suitable for studying the biomechanics of this dynamic aggregation process. Mechanobiology has been instrumental in advancing our understanding of infectious diseases, cancer biology, developmental biology, and other fields. There is still tremendous potential in exploring this area of research further in the fields of protein aggregation and neurodegeneration, particularly with intelligent approaches that adapt in real time to optimally retrieve the mechanical properties. This could help us uncover new insight into the mechanisms by which the process of protein aggregation and inclusion formation alters the mechanical properties of cells, their physiological properties, and potentially disrupts their function. Changes in the mechanical properties of cells and protein aggregates could also influence rates or mechanisms of aggregate clearance. A better understanding of the relationship and functional consequences of protein-aggregation-dependent changes in mechanical properties of cells could pave the way for new experimental approaches, research directions, and therapeutic strategies.

Our methods can enable a better understanding of the underlying mechanisms that occur in neurodegenerative diseases. For example, IC-LINA can be used to detect aggregates without the need for labels, offering higher fidelity information for high-throughput screens that normally rely on fluorescent labeling. It could be used to test the effect of drug candidates or small molecules on the propensity of aggregation, including their size, morphology, and other properties, and viscoelasticity, if combined with other modalities like BM. Currently, there are limited tools that enable investigating the interactions between drug candidates and NDD-associated aggregating proteins in the early stages of aggregate formation, oligomerization. The aggregation onset prediction model, AEGON, can be used to automatically follow the process in multiple modalities with high spatiotemporal resolution and study the dynamic interactions between the aggregating protein and drug candidates or other proteins, organelles, or subcellular structures. SDM has exciting potential to impact the field of neurodegenerative diseases and other amyloid-associated diseases, enabling scientists to automate tedious tasks and focus on data analysis and interpretation, and to open up new possibilities for experiments that were impossible beforehand. Aside from leading to important contributions, SDM can be extended in the future to identify interesting events in an unsupervised manner to discover unknown biological phenomena and open up new avenues of research.

## Methods

### Sample preparation

HEK293 cells were cultured at 37 °C and 5% $CO_2$ using DMEM high glucose without phenol red (Gibco, Thermo Fisher Scientific), supplemented with 10% fetal bovine serum, 1% penicillin-streptomycin, and 4 mM L-glutamine (all three from Gibco, Thermo Fisher Scientific). Cells were plated at a density of 120,000 per dish on FluoroDish Sterile Culture Dishes 35 mm, 23 mm well (World Precision Instruments), coated with fibronectin. Cells were transfected one day after plating using polyethylenimine (PEI) transfection. 2 μg of DNA is mixed in 100 μl of OptiMEM Reduced-Serum Medium (Life Technologies), 6 μl of PEI is mixed in 100 μl of OptiMEM, and then both mixtures are mixed and incubated for 5 min at room temperature (RT), then added dropwise and carefully distributed over the cells. Cells were then returned to the incubator and left there until imaging was initiated.

### Data acquisition and processing

For our standard imaging experiments, it was performed with a custom-built microscope equipped with a temperature and $CO_2$ controlled incubator for live cell imaging, as described in previous work from our group[43]. Live-cell imaging was performed in DMEM without phenol red at 37 °C and 5% $CO_2$. The microscope is controlled using custom LabVIEW software. For fluorescence imaging, a 120 mW, 488 nm laser (iBeam smart, Toptica), is focused into the back focal plane of an Olympus UPLSAPO 60XW 1.2 NA objective for wide-field epi-fluorescence illumination. The fluorescence light was filtered using a combination of a dichroic mirror (zt405/488/532/640/730rpc, Chroma) and an emission filter. For phase imaging, we used the white-light Koehler illumination module of a Zeiss Axiovert 100 M microscope equipped with a halogen lamp to collect brightfield images, which are later processed into QPI images. The detection path is arranged as a sequence of four 2-f configurations to provide image–object space telecentricity. The image splitter placed behind the last lens directs the light into eight images, which are registered by two synchronized sCMOS cameras (ORCA Flash 4.0, Hamamatsu; back-projected pixel size of 111 nm). For translating the sample, the microscope is equipped with piezoLEGS stage (3-PT-60- F2,5/5) and Motion-Commander-Piezo controller (Nanos Instruments GmbH).

We used custom MATLAB (Mathworks) scripts (available here) to retrieve the phase information from the brightfield images and produce quantitative phase images. These scripts are also used for pixel-registration in the 8 z-planes for both phase and fluorescence images. The images are cropped to a size of 352 pixels × 352 pixels. Python was used for data analysis and plotting.

### Brillouin microscopy

The BM setup is composed by a commercial Zeiss body (Axiovert 200 M) coupled with a custom-built two-stage VIPA spectrometer[54], with the addition of a Lyot stop to increase the suppression of the elastically scattered light[55]. The laser used for Brillouin imaging has a wavelength of 660 nm (Torus 660, Coherent), and the power on the sample was kept below 7 mW. The exposure time for a single point was set to 100 ms. In the assumption of a Lorentzian lineshape, the

instrumental response of the spectrometer (270 MHz) was subtracted from the measured linewidth to deconvolve it. The objective used for imaging was a 40 × 1.0 NA objective. The LabVIEW software that controls the microscope was adjusted to interface with Python and run DL models in real time. The user is able to set the parameters for the initial and the optimized scan.

To extract the Brillouin shift and linewidth of the aggregating regions and the periphery, we performed thresholding (Otsu's method) on the fluorescence, Brillouin shift, and Brillouin linewidth images to produce masks, which we then used to segment the corresponding regions. The fluorescence image is used to localize the aggregating region. The shift and linewidth images are used to obtain the full region in which we have signal, without the background—and the aggregating region that is obtained from fluorescence masks is subtracted to obtain only the periphery region. The fluorescence image is used to localize the aggregating region. The shift and linewidth images are used to obtain the full region in which we have signal, without the background—and the aggregating region that is obtained from fluorescence masks is subtracted to obtain only the periphery region.

### Neural network training
We have trained neural network models of different architectures, namely, VGG16-LSTM, ViViT, and their single-time-point counterparts (VGG16 and ViT). The full dataset consists of 71 events and 69 non-events. We used TensorFlow and Keras to build our models, and training was done on a workstation equipped with an NVIDIA GeForce RTX 3090 GPU. We used the adaptive moment estimation (Adam) optimizer with a learning rate of 1e-5 and a binary cross-entropy loss function. Before training our models, we normalize both the phase and fluorescence images by rescaling each image to be between 0 and 1. We used 15% of our dataset as the test set and split the rest of the dataset into training and validation sets, with 20% being used for validation. We used the "EarlyStopping" (on the validation loss) and "ModelCheckpoint" callbacks to avoid overfitting and to save the best, most general models. The time taken to train a successful model and for inference depends on the computing capabilities available to the user. On our machine with the RTX 3090, training a successful model can be done in a range of a few minutes, and the model can do the inference in 60 ms. We did not observe a noticeable difference in training or inference times between our different models.

### Self-driving microscopy
We customized our LabVIEW software (which will be made available upon publication) to develop our SDM pipeline. The graphical user interface (GUI) is shown in Supplementary Fig. 4. The process begins with an initial scan, after which the model assesses the likelihood of upcoming events. Based on these predictions, the microscope autonomously adjusts imaging settings and launches an optimized scan. The software is highly modular, allowing users to configure a variety of parameters for both the initial and optimized scans, such as imaging modalities, exposure times, and laser powers. After the initial scan, images are given in real time as input to the model, which predicts whether aggregation is about to occur. There are two modes, one where a specific acquisition sequence (a patterned scan in both the x and y directions) is performed and all the images are passed to the model together at the end, and one where the images at each FOV are passed directly to the model. In the first mode, all FOVs are checked at the same time, and if there are any aggregation events predicted, they are added to a list, which is iterated over in the optimized scan. In the other mode, once an event is predicted, this directly initiates the optimized scan at that FOV, after which the initial scan continues to check the subsequent FOVs.

### Reporting summary
Further information on research design is available in the Nature Portfolio Reporting Summary linked to this article.

## Data availability
The training, validation, and test datasets are available on Zenodo. Source data are provided. Source data are provided with this paper.

## Code availability
The Python codes for training a model and using/testing a pre-trained model, along with the ViT models (with 1 plane and 8 planes inputs), are available on GitHub, and the software used for self-driving microscopy is available on Zenodo.

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

## Acknowledgements

K.A.I., L.F. and A.R. were supported by funding through the Swiss National Science Foundation (SNSF) through the National Centre of Competence in Research Bio-Inspired Materials and SNF grant CRSII5_193740. R.P. acknowledges support of an ERC Consolidator Grant (no. 864027, Brillouin4Life), the German Center for Lung Research (DZL), as well as research funding 'Life Science' of the Molit Institute. This work was supported by funds from the European Molecular Biology Laboratory.

## Author contributions

K.A.I. conceived the project; K.A.I., H.A.L. and A.R. designed the experiments; K.A.I., L.F. and C.B. prepared the samples; K.A.I. performed the experiments, acquired, analyzed, curated and visualized the data; K.A.I. and C.C. developed and optimized the neural network training pipeline; K.A.I. designed the SDM software pipeline, implemented the codes to be run in real time and performed the SDM experiments using the trained models; C.B. and R.P. designed and developed the BM setup; C.B. assisted in BM experiments and designed and implemented the SDM software in the BM setup, with design input from K.A.I.; K.A.I. designed the figures and wrote the manuscript, with input from all authors; K.A.I., R.P., H.A.L. and A.R. supervised the project; All authors discussed the results and commented on the manuscript.

## Competing interests

H.A.L. has received funding from the industry to support research on neurodegenerative diseases, including from Merck Serono, UCB and AbbVie. These companies had no specific role in the conceptualization, preparation, or decision to publish this work. H.A.L. is also the co-founder and Chief Scientific Officer of ND BioSciences SA, a

company that develops diagnostics and treatments for neurodegenerative diseases based on platforms that reproduce the complexity and diversity of proteins implicated in neurodegenerative diseases and their pathologies. All remaining authors declare no competing interests.
