## [Transparent Peer Review file · Nature Communications]

Self-Driving Microscopy Detects the Onset of Protein Aggregation and Enables Intelligent Brillouin Imaging

Corresponding Author: Professor Aleksandra Radenovic

Version 0:

Reviewer comments:

Reviewer #1

(Remarks to the Author)

The manuscript by Ibrahim et al. presents an AI-assisted microscope designed to detect and localize the formation of protein aggregates. The authors report an accuracy of 91% for their model, trained using both fluorescence and unstained QPI images, and demonstrate a combined Brillouin imaging modality to investigate the mechanical properties of rapidly forming aggregates. The study of phase transitions in protein aggregates is indeed an important topic, and Brillouin microscopy has the potential to contribute meaningfully to this field. However, the present manuscript lacks convincing evidence to support the effectiveness of the proposed method:

- The authors claim an accuracy of 96% for the LINA (IC-LINA) model in detecting aggregates without fluorescent labels, which is surprisingly higher than the 91% accuracy obtained using labeled fluorescent images. The authors should elaborate on this finding, providing additional evidence of their system's ability to detect aggregates using unstained samples alone.
- The extension of the field of view in which the detection and localization of aggregates occur with the declared precision must be reported.
- The authors claim an increased Brillouin shift for the protein aggregates. However, this is not evident in Fig. 3. It would be helpful to mark the regions where aggregates are expected to form in the associated QPI and Brillouin images, possibly through image segmentation with the fluorescent signal. Furthermore, the manuscript lacks reported values for the measured Brillouin shift and linewidth of the aggregates.
- Associated Brillouin linewidth maps are shown in the supplementary material. Given that viscosity is a critical parameter in determining liquid-to-solid phase transitions in protein aggregates, including such frames in the main text and comparing them with the reported Brillouin frequency shift would add value.
- The pixel dwell time for acquiring single Brillouin spectra was reported to be 100 ms, while Brillouin images were acquired every 5 minutes to track aggregate evolution. Given the typical micron-scale size of the aggregates, it is important to specify the number of Brillouin spectra acquired in the illustrated maps and the associated scanning step size to evaluate the resolution capability of this method for monitoring aggregate formation.
- Fig. 4 is difficult to evaluate, as the aggregate shown in the Brillouin map appears much larger than the one marked with the fluorescent label. Since the aggregate localization ability of the intelligent microscope using unstained images represents the manuscript's major claim, further data supporting this claim should be collected. This could include showing a better correspondence between the identified aggregates in QPI and the associated Brillouin images. The authors may also consider using additional fluorescent markers to provide a clearer visualization of the selected region of interest.

Reviewer #2

(Remarks to the Author)

This paper by Ibrahim et. al. presents a smart microscopy approach for the automated detection of protein aggregation. They train deep learning models based on different architecture to detect when protein aggregation is occurring. This automated detection, in turn, enables live event-triggered imaging pipelines e.g. switching microscopy modality to acquire Brillouin scattering images of the aggregates. Importantly, the authors test models trained with increasingly simplified inputs (full 3D timelapses, individual stacks and individual images) and find that even the simplest data can give a high detection accuracy of 86%. The model comparison they provide and the resources shared in the paper (code, training data) will be very helpful

to other researchers interested in studying protein aggregation with a high-throughput. Overall, this self-driving microscopy method is an exciting advance and will be of inspiration to many researchers, however I have a few recommendations to improve the clarity and reproducibility of the work (see below).

Major points:

-Do Httex1 aggregates always form in a single focal spot per cell / field of view? Would this affect model predictions? This could be a good point to address in the text / discussion, as it might influence applicability of the method.

-The comparison of different models (architectures, inputs) is very helpful but it could benefit from further details being included in the text/ methods. For example, a clear indication of the total sample size used to train/test the different models and an estimated time required for training / prediction would be extremely informative for anyone wanting to apply this method. It would also be interesting to know whether the models perform any differently in terms of prediction time. Overall this will help other researchers assess the resources needed for potential re-training or fine-tuning of the existing models.

-Data availability: in addition to the training, validation and test datasets I would recommend the authors also share the trained models. Two models are included in an online folder but this is not mentioned in the text and I would recommend sharing all models developed in this work, including the ones that work on brightfield images.

-Some of the statements on using Brillouin microscopy to detect protein aggregates would need additional support/clarification. For example: "Furthermore, we demonstrate that Brillouin microscopy can circumvent the need for fluorescence labeling to identify mature protein aggregates." However, looking at main text Figure 3d, hardly any change is visible in the Brillouin shift whereas an aggregate is clearly visible in the fluorescence image at time point = 10 min. This is clearer in the timelapse of the Brillouin linewidths (Supp Fig 3), which I would suggest moving to the main text figure. Could the authors also comment on whether Brillouin readouts relate to the material properties of aggregates, for example do they differ between liquid-liquid droplets that recover in FRET compared to more solid and static aggregates? A more detailed discussion of this with references in the text would be helpful.

Minor points:

-Some of the figure fonts are too small, e.g. Fig 1c T1, T2, Figure 2c, Figure 3 axes labels and Brillouin scale.

-It could be interesting to extend the discussion and for the authors to comment on the applicability of this method and model to areas beyond protein aggregation. Perhaps it could be relevant also for assays based on protein relocalisation, eg to detect nuclear accumulation?

Version 1:

Reviewer comments:

Reviewer #2

(Remarks to the Author)

The revised manuscript by Ibrahim et. al. has been improved with more details and clarifications and all reviewers' concerns have been addressed, therefore I recommend the paper for publication.

Self-Driving Microscopy Detects the Onset of Protein Aggregation and Enables Intelligent Brillouin Imaging

Response to Reviewers

We are grateful to the reviewers for their careful reading of the manuscript and their constructive feedback and recommendations, all of which have contributed significantly to improving the quality and impact of the manuscript.

We have made every effort to address their questions and provide thorough responses to their requests for clarification and their inquiries. Please find enclosed a point-by-point response to all their questions and inquiries.

Reviewer comments are reproduced in black, **our answers are in blue**, and **references to the manuscript are in green**. Citations correspond to a reference list at the end of this document. Please note that all line and page numbers refer to the revised manuscript.

Reviewer #1:

The manuscript by Ibrahim et al. presents an AI-assisted microscope designed to detect and localize the formation of protein aggregates. The authors report an accuracy of 91% for their model, trained using both fluorescence and unstained QPI images, and demonstrate a combined Brillouin imaging modality to investigate the mechanical properties of rapidly forming aggregates. The study of phase transitions in protein aggregates is indeed an important topic, and Brillouin microscopy has the potential to contribute meaningfully to this field. However, the present manuscript lacks convincing evidence to support the effectiveness of the proposed method:

We thank the reviewer for highlighting the importance of our work and the significance of developing more advanced Brillouin microscopy tools to study protein aggregation.

- The authors claim an accuracy of 96% for the LINA (IC-LINA) model in detecting aggregates without fluorescent labels, which is surprisingly higher than the 91% accuracy obtained using labeled fluorescent images. The authors should elaborate on this finding, providing additional evidence of their system's ability to detect aggregates using unstained samples alone.

We would like to clarify that the difference in accuracy arises because the two models are designed to perform distinct tasks, each with a different level of difficulty.

- 1- The label-free model (IC-LINA) is trained to determine **whether a mature aggregate is present in an image**. In our previous work¹, we showed that it is possible to localize and segment aggregates **purely from label-free images**, and we characterized and validated this approach in depth, showing that it is highly reliable. Here, we build on our previous work¹; instead of segmenting and localizing aggregates (pixel classification), the task is simplified: the model informs us whether an aggregate exists in this FOV or not (image classification).

This task would be straightforward to do with fluorescence images, which show the Httex1 protein with high specificity, as mature aggregates are distinct from soluble protein—they are much more densely packed, globular, smaller in size, and much higher in intensity.

- 2- The fluorescence-based model is trained to predict whether **the onset of aggregation is about to occur**, i.e., predicting aggregation before it happens – from images of soluble protein. This is a much more challenging task because it requires detecting subtle, pre-aggregative changes in soluble protein distributions, which are not visually discernible to the human eye and have not been previously predicted in the field.

Thus, while the fluorescence-based model uses high-contrast and high-specificity fluorescence images, its significantly harder task results in a lower accuracy. If we trained a model to detect mature aggregates using fluorescence images (rather than predicting their formation), it would likely achieve nearly 100% accuracy, as fluorescence labeling provides a highly specific and unambiguous signal compared to brightfield or QPI images.

In summary, although the 2nd model uses fluorescence inputs, it has a much harder task (i.e., predicting aggregation onset) to do and therefore has a lower accuracy.

We have now named the fluorescence-based model for predicting aggregation onset (AEGON) and refer to it in the manuscript to distinguish it from IC-LINA and improve clarity / avoid potential confusion.

- The extension of the field of view in which the detection and localization of aggregates occur with the declared precision must be reported.

We have clarified the field of view and precision information in the revised manuscript (lines 179, 239, and 314). The added lines are copied below:

The FOV of the input images is $\approx 39 \mu\text{m} \times 39 \mu\text{m}$.

The FOVs in the images are $35 \mu\text{m} \times 35 \mu\text{m}$ (with a step size of $1 \mu\text{m}$ for BM images).

The FOV of the input image (and GT fluorescence images) is $\approx 39 \mu\text{m} \times 39 \mu\text{m}$, and the Brillouin image focuses on the aggregate region (FOV = $10 \mu\text{m} \times 10 \mu\text{m}$ with a step size of $0.2 \mu\text{m}$).

- The authors claim an increased Brillouin shift for the protein aggregates. However, this is not evident in Fig. 3. It would be helpful to mark the regions where aggregates are expected to form in the associated QPI and Brillouin images, possibly through image segmentation with the fluorescent signal. Furthermore, the manuscript lacks reported values for the measured Brillouin shift and linewidth of the aggregates.

We now demonstrate this in further depth in the revised manuscript. We have marked in Figure 3 the regions where aggregates are, based on the fluorescence signal. The new Figure 4 (shown below) contains the linewidth images (as suggested by both reviewers) and a comparison of the Brillouin shifts and linewidths of the aggregate regions and the periphery (cytoplasm). We also report the final values of the measured Brillouin shift and linewidth in the main text.

Figure 4: Evolution and comparison of the Brillouin shift and linewidth over time. (a) Dynamic imaging of the linewidths throughout aggregation is enabled by the model which predicts the onset of the process. Scale bar: $5 \mu\text{m}$. (b, c) Comparison of the Brillouin shifts and linewidths of the aggregating region versus the periphery. TP – timepoint. (b) Brillouin shift comparison. (c) Linewidth comparison.

To extract the Brillouin shift and linewidth of the aggregating regions and the periphery, we performed thresholding (Otsu's method) on the fluorescence, Brillouin shift, and Brillouin linewidth images to produce masks, which we then used to segment the corresponding regions. The fluorescence image is used to localize the aggregating region. The shift and linewidth images are used to obtain the full region in which we have signal, without the background – and the aggregating region that is obtained from fluorescence masks is subtracted to obtain only the periphery region. We describe this in the methods section.

- Associated Brillouin linewidth maps are shown in the supplementary material. Given that viscosity is a critical parameter in determining liquid-to-solid phase transitions in protein aggregates, including such frames in the main text and comparing them with the reported Brillouin frequency shift would add value.

We agree and have added the linewidth figure to the main text as Figure 4 (and Figure 4 has become Figure 5).

- The pixel dwell time for acquiring single Brillouin spectra was reported to be 100 ms, while Brillouin images were acquired every 5 minutes to track aggregate evolution. Given the typical micron-scale size of the aggregates, it is important to specify the number of Brillouin spectra acquired in the illustrated maps and the associated scanning step size to evaluate the resolution capability of this method for monitoring aggregate formation.

We have clarified the FOV sizes and scanning step sizes in the revised manuscript (lines 239 and 314), so the number of spectra can be simply calculated. The FOV and step size / resolution of the scanned images is definable by the user (limited by the acoustic phonon size). There is a trade-off between FOV size, resolution, imaging speed, and sample health. In our experiments, we set a frame rate of 1 image every 5 minutes to reduce the light dose brought upon the sample, which could influence the process and perturb the aggregation / prevent it from occurring. Since it is a living specimen and a dynamic process, this limits the choice of the FOV size and resolution, to have a fast enough imaging rate that does not miss the process, while also not perturbing it with high laser irradiances for too long. It could be possible to image even faster and with higher resolution. This remains to be explored in future studies.

- Fig. 4 is difficult to evaluate, as the aggregate shown in the Brillouin map appears much larger than the one marked with the fluorescent label. Since the aggregate localization ability of the intelligent microscope using unstained images represents the manuscript's major claim, further data supporting this claim should be collected. This could include showing a better correspondence between the identified aggregates in QPI and the associated Brillouin images. The authors may also consider using additional fluorescent markers to provide a clearer visualization of the selected region of interest.

We would like to clarify that this perceived larger size is due to the different scales of the images. The brightfield and fluorescence images are $\approx 39 \mu\text{m} \times 39 \mu\text{m}$, while the Brillouin image focuses on the aggregate region and has a size of $10 \mu\text{m} \times 10 \mu\text{m}$, due to the slow imaging speed of BM and the high-precision (step size of $0.2 \mu\text{m}$) that we chose to use to get as much detail as possible. The images have different scale bars (both correspond to $5 \mu\text{m}$, but the BM scale bar is larger to account for the smaller FOV). We have added information on the FOVs as previously suggested by the reviewer; we believe this will help clarify the difference in scale to readers of the manuscript.

Reviewer #2:

This paper by Ibrahim et. al. presents a smart microscopy approach for the automated detection of protein aggregation. They train deep learning models based on different architecture to detect when protein aggregation is occurring. This automated detection, in turn, enables live event-triggered imaging pipelines e.g. switching microscopy modality to acquire Brillouin scattering images of the aggregates. Importantly, the authors test models trained with increasingly simplified inputs (full 3D timelapses, individual stacks and individual images) and find that even the simplest data can give a high detection accuracy of 86%. The model comparison they provide and the resources shared in the paper (code, training data) will be very helpful to other researchers interested in studying protein aggregation with a high-throughput. Overall, this self-driving microscopy method is an exciting advance and will be of inspiration to many researchers, however I have a few recommendations to improve the clarity and reproducibility of the work (see below).

We thank the reviewer for their positive assessment of our work and appreciate their kind words.

Major points:

-Do Httex1 aggregates always form in a single focal spot per cell / field of view? Would this affect model predictions? This could be a good point to address in the text / discussion, as it might influence applicability of the method.

Httex1 aggregates generally form in one single spot per cell. If there are multiple cells in the FOV, it is possible that multiple aggregates form in the same FOV; however, this rarely occurs due to the stochastic nature of the process and the low likelihood of multiple cells aggregating at the same time and place. When we were collecting our training dataset, we tried to include as many different kinds of examples as possible to produce the most generalizable and accurate model we can. We did not exclude any kinds of aggregation examples that we collected.

We agree that this is an interesting point, and we now discuss it in the revised manuscript, starting from line 180 (copied below):

Generally, the soluble protein aggregates into only one spot during the process. If an FOV contains multiple cells, it is possible that each cell could form a separate aggregate; however, this rarely occurs as the likelihood of two cells forming aggregates at the same time and place is low. During our collection of training events, we aimed to collect a diverse dataset that is well-representative of the different phenomena that could occur and did not exclude any particular kinds of aggregation. The model should, therefore, be able to capture varying types of aggregation events. Further fine tuning of the model is also possible using transfer learning.

-The comparison of different models (architectures, inputs) is very helpful but it could benefit from further details being included in the text/ methods. For example, a clear indication of the total sample size used to train/test the different models and an estimated time required for training / prediction would be extremely informative for anyone wanting to apply this method. It would also be interesting to know whether the models perform any differently in terms of prediction time. Overall this will help other researchers assess the resources needed for potential re-training or fine-tuning of the existing models.

We thank the reviewer for this suggestion and have included more information in the revised manuscript (lines 165, 500, and 509), copied below.

The full dataset consists of 71 events and 68 non-events.

The time taken to train a successful model and for inference depends on the computing capabilities available to the user. On our machine with the RTX 3090, training a successful model can be done in a range of a few minutes, and the model can do the inference in 60 ms. We did not observe a noticeable difference in training or inference times between our different models.

-Data availability: in addition to the training, validation and test datasets I would recommend the authors also share the trained models. Two models are included in an online folder but this is not mentioned in the text and I would recommend sharing all models developed in this work, including the ones that work on brightfield images.

We agree. All the models we have available will be made accessible online, along with the data and the code, on GitHub and Zenodo repositories, that will be available upon publication.

-Some of the statements on using Brillouin microscopy to detect protein aggregates would need additional support/clarification. For example: “Furthermore, we demonstrate that Brillouin microscopy can circumvent the need for fluorescence labeling to identify mature protein aggregates.” However, looking at main text Figure 3d, hardly any change is visible in the Brillouin shift whereas an aggregate is clearly visible in the fluorescence image at time point = 10 min. This is clearer in the timelapse of the Brillouin linewidths (Supp Fig 3), which I would suggest moving to the main text figure. Could the authors also comment on whether Brillouin readouts relate to the material properties of aggregates, for example do they differ between liquid-liquid droplets that recover in FRET compared to more solid and static aggregates? A more detailed discussion of this with references in the text would be helpful.

We have moved the figure of the linewidths to the main text, as suggested by both reviewers. We now also expand our analysis and show that the aggregating region has a higher Brillouin shift and linewidth than the periphery (Figure 4).

We have amended the sentence (line 25) and combined it with the sentence that previously came after it, to minimize confusion and to clarify our message – that fluorescence labeling, which is what has usually been used to classify FOVs that have aggregates, is not necessary, as we show that it is possible to classify FOVs purely from label-free brightfield images using a neural network. The new sentence is copied below:

Furthermore, we demonstrate that by detecting mature aggregates in real time using brightfield images and a neural network, Brillouin microscopy can be used to study their biomechanical properties without the need for fluorescence labeling, minimizing phototoxicity and preserving sample health.

We have commented on the point raised by the reviewer and added a discussion paragraph, starting at line 381 of the revised manuscript.

BM provides insight into the high-frequency longitudinal modulus of biological materials and is therefore sensitive to their viscoelastic properties. Prior work has shown that Brillouin shifts increase in more solid-like structures, such as mature Httex1 aggregates, compared to the surrounding cytoplasm³⁶. This has also been shown for biomolecular condensates formed via liquid-liquid phase separation (LLPS), such as stress granules that are formed through the expression of FUS and the presence of oxidative stress³⁵. Recent work uses BM as a measure for the average molecular interaction and finds that conditions that favor LLPS increase the Brillouin shift, linewidth, and protein

concentration⁵². We expect that the stiffness of aggregating proteins will increase as they transition from liquid-like states to the solid-like mature-aggregate state, due to higher protein density and stronger intermolecular interactions that reduce mobility and increase mechanical rigidity. Our SDM-Brillouin framework is well suited for exploring a continuum of material states throughout the aggregation process, including transient, dynamic condensates, as it enables dynamic imaging throughout the process for the first time, and as shown here we observe an increase in the Brillouin shift as the aggregate forms. Future work which incorporates additional modalities, such as fluorescence recovery after photobleaching (FRAP), may allow for a more systematic correlation between internal dynamics and Brillouin mechanical signatures, shedding light on the phase behavior and maturation of aggregates in neurodegenerative disease models.

Minor points:

-Some of the figure fonts are too small, e.g. Fig 1c T1, T2, Figure 2c, Figure 3 axes labels and Brillouin scale.

We have enlarged the fonts.

-It could be interesting to extend the discussion and for the authors to comment on the applicability of this method and model to areas beyond protein aggregation. Perhaps it could be relevant also for assays based on protein relocalisation, eg to detect nuclear accumulation?

We thank the reviewer for the suggestion and have added a paragraph discussing this in line 327.

While this work focuses on applying SDM and intelligent BM to protein aggregation, the underlying strategy of real-time event detection and adaptive multimodal imaging is broadly applicable to a variety of dynamic cellular processes. For instance, many high-content screening assays rely on protein re-localization events — such as nuclear translocation of transcription factors, signaling molecules, or stress response proteins — as key readouts. Our SDM framework, which can be trained to detect subtle changes from single-plane fluorescence or brightfield images, could be adapted to identify such re-localization events, including nuclear accumulation, cytoplasmic clearance, or organelle-specific targeting. This would enable dynamic switching to higher-resolution or complementary modalities (e.g., super-resolution or label-free techniques), or optimizing other imaging settings such as the temporal resolution or the exposure time/laser power, at the right time and location, enhancing the informativeness of such assays while minimizing phototoxicity and enhancing throughput. We envision that retraining the model on datasets relevant to nuclear import/export, mitotic events, or phase transitions could allow for real-time monitoring of diverse biological phenomena across cell biology, developmental systems, or drug screening applications.

References

1. Ibrahim, K. A. *et al.* Label-free identification of protein aggregates using deep learning. *Nat Commun* **14**, 7816 (2023).